# Impact of Climate Conditions on Pollutant Concentrations in the Effluent from a One-Stage Constructed Wetland: A Case Study

Agnieszka Operacz [1,*], Krzysztof Jóźwiakowski [2], Joanna Rodziewicz [3], Wojciech Janczukowicz [3] and Piotr Bugajski [1]

1 Department of Sanitary Engineering and Water Management, University of Agriculture in Krakow, Mickiewicza Av. 21, 31-120 Krakow, Poland; p.bugajski@urk.edu.pl
2 Department of Environmental Engineering and Geodesy, University of Life Sciences in Lublin, Leszczyńskiego St. 7, 20-069 Lublin, Poland; krzysztof.jozwiakowski@up.lublin.pl
3 Department of Environment Engineering, University of Warmia and Mazury in Olsztyn, Warszawska St. 117a, 10-719 Olsztyn, Poland; joanna.rodziewicz@uwm.edu.pl (J.R.); jawoj@uwm.edu.pl (W.J.)
* Correspondence: a.operacz@urk.edu.pl

**Abstract:** This paper reports the results of an investigation into the influence of precipitation and air temperature on the efficiency of pollutant removal processes and effluent pollutant concentrations in a one-stage constructed wetland with subsurface vertical flow. We studied an on-site constructed wetland system that used *Phragmites australis* for the treatment of domestic wastewater. The system was located in central Europe, in the south-east of Poland, in a temperate climate zone with transitional features. Physico-chemical analyses of influent and effluent wastewater, as well as measurements of precipitation and air temperature were carried out in the years 2001–2010. It was shown that the pollutant removal efficiency of the treatment plant was significantly higher in the growing season than outside the growing season (the mean efficiency is usually a few percent higher but generally this parameter is highly varied). This indicated that temperature determined the efficiency of the wastewater treatment. We found that the amount of precipitation affected the concentration of pollutants in the effluent. The more rainfall there was, the lower the content of pollutants in the effluent from the treatment plant, which demonstrated that rainwater diluted the concentrations of pollutants in the treated wastewater—thus improving the efficiency of the wastewater treatment plant.

**Keywords:** one-stage constructed wetland system; wastewater treatment; precipitation; temperature

## 1. Introduction

In recent years, ongoing climate change has gained an unprecedented pace. The rapid alterations that are occurring have motivated people to take measures to protect the environment and to solve environmental issues, one of the most urgent of which is the scarcity or lack of water [1–3]. At the same time, the development of housing, which is an inextricable component of the development of civilization, has increased the amount of water used for domestic purposes, as well as for watering animals and agricultural crops [4]. Of course, the more water people use for living purposes, the more wastewater they produce, which—when untreated or improperly treated—threatens the already scarce water resources of many countries, including European ones [5], and especially countries that are at risk of drought [6]. Currently, one of the most important tasks of decision makers, urban planners, environmental engineers, and designers is to consolidate their activities so that the development of housing can go hand in hand with preventing the degradation of the natural environment. A common cause of degradation of the aquatic environment of rivers, lakes, and groundwater reservoirs is the improper treatment of wastewater that is discharged into water bodies from human settlements. A notorious example is the

contamination, in July 2022, of the Oder River, which is one of the largest rivers in Poland and Europe. This example is a vivid reminder of the fact that professional sanitation infrastructure should be built wherever housing development takes place. It is vital that the construction of new wastewater treatment plants or the modernization of already existing ones should be carried out using high-efficiency wastewater treatment technologies, and that these facilities should be properly maintained and controlled during their period of use. In the case of collective treatment plants located in large urban agglomerations, treatment processes are monitored non-stop, with the operator being able to react quickly to adverse events (such as the release of toxic sewage into the plant) [7,8]. Many small collective treatment plants, mostly those located in rural areas, are not continuously monitored. In these type of facilities, operational problems that are not noticed and resolved within a short time create the risk of contamination of the waters of the receiving body with excessive amounts of pollutants—pollutants that have not been removed in the treatment process. The control of the processes taking place in on-site domestic wastewater treatment plants is even poorer, since the legislation in some countries, such as Poland, does not call for the quality control of wastewater that is discharged from those facilities into surface waters or into the ground [9]. As the literature data clearly show, many such facilities do not operate properly, i.e., the processes taking place in the technological line do not guarantee the efficient removal of pollutants from wastewater [10,11]. Given the specific manner in which domestic wastewater treatment plants operate, they should have an uncomplicated structure and be characterized by a high wastewater treatment efficiency. Off-grid sewage systems for single-family homes (other than cesspools [12]) most often consist of a septic tank with a soakaway system for infiltrating wastewater into the ground (a drainfield), or containerized plants with a biological treatment unit that use the activated sludge process. Neither of these two types of systems guarantee the efficient removal of pollutants from wastewater [10,13–15]. Therefore, dwellings that do not have, and are not expected to have, appropriate control measures in place for the continuous monitoring of treatment processes should be fitted with systems that have a simple structure, do not require constant operational supervision, and are robust to unsteady inflows of sewage. One type of such systems, which can be used either as on-site domestic wastewater treatment plants or as collective treatment plants, are constructed wetlands (CWs) [16–18]. CWs are highly reliable in removing both organic and biogenic pollutants [19]. Since, in this type of facility, the basic site of biological treatment processes is the plant–soil bed, it is important to determine whether these processes can work properly in the climatic conditions of a given geographical region. According to research reports, constructed wetlands show good treatment efficiency in changing hydraulic and climatic conditions. Therefore, in this paper, we put forward a thesis that the climatic conditions of southeastern Poland affect the efficiency of the removal of organic and biogenic pollutants in a constructed wetland.

The aim of the present study was to determine the effect of air temperature and the amount of precipitation on the concentrations of organic and biogenic pollutants in wastewater effluent from an investigated CW, which services a school building in Sobieszyn, Province of Lublin, Poland. Air temperature is a factor that has a significant impact on the vegetation of plants that play a key role in removing pollutants in CWs. Low air temperature in winter results in the disappearance of vegetation, and thus the "outgrowing period" is a period of expected lower efficiency for the treatment plant. In other words, air temperature exerts its greatest influence indirectly by providing optimal conditions for plant growth, or by blocking or causing the complete disappearance of vegetation. Thus, it can be concluded that temperature is the main factor that determines the vegetation of plants in CWs. Ji et al. [20] indicated that the biochemical and microbiological processes in CW beds work properly at air temperatures above 5 °C. Studies [21–23] have shown that the nitrogen removal processes from wastewater are more efficient in warmer periods of the year. Mietto et al. [24] observed a linear relationship between air temperature fluctuations and the efficiency of the nitrification and removal of nitrogen compounds in a VF–HF hybrid system in northern Italy. It was found that the nitrogen reduction rate was clearly

lower in the winter months (January, February, and March). On the other hand, research conducted in hybrid CWs in Polish conditions [25] did not show a statistically significant impact of low air temperatures in the winter season on the effectiveness of reducing $BOD_5$ and COD values, as well as in terms of removing TSS. The temperature mainly affects the vegetation intensity of the plants planted in the CWs. Nevertheless, there may be extreme conditions that disturb the functioning of these systems. For example, too low a temperature (e.g., below $-10\ ^{\circ}C$) in the non-growing season may cause a periodic freezing of the surface of vertical-flow beds, which may block the outflow of sewage. On the other hand, too high temperatures may intensify evaporation and may lead to the lack of sewage outflow from the deposits [26]. However, no such extreme situations have been identified in this investigation; therefore, these potential phenomena have not been included in the advanced analysis.

CW systems are much more exposed to rainwater inflow compared to wastewater treatment plants with activated sludge or a biological bed (which are usually underground or covered structures). High precipitation increases the hydraulic load of a CW bed, with large quantities of rainwater diluting the wastewater flowing through the bed. Precipitation rates in countries with moderate climates (including Poland) ranges from 500 to 800 mm/year (Figure 1). Descriptive studies determining the technological reliability of CWs in removing pollutants in different seasons (depending on seasonal temperature changes) can be found in the literature, and they are devoted to the operation of CWs under climatic conditions similar to Polish ones [27,28]. By contrast, research reports on the influence of the amount of precipitation on the concentration of pollutants in wastewater effluent from CW systems are scarce.

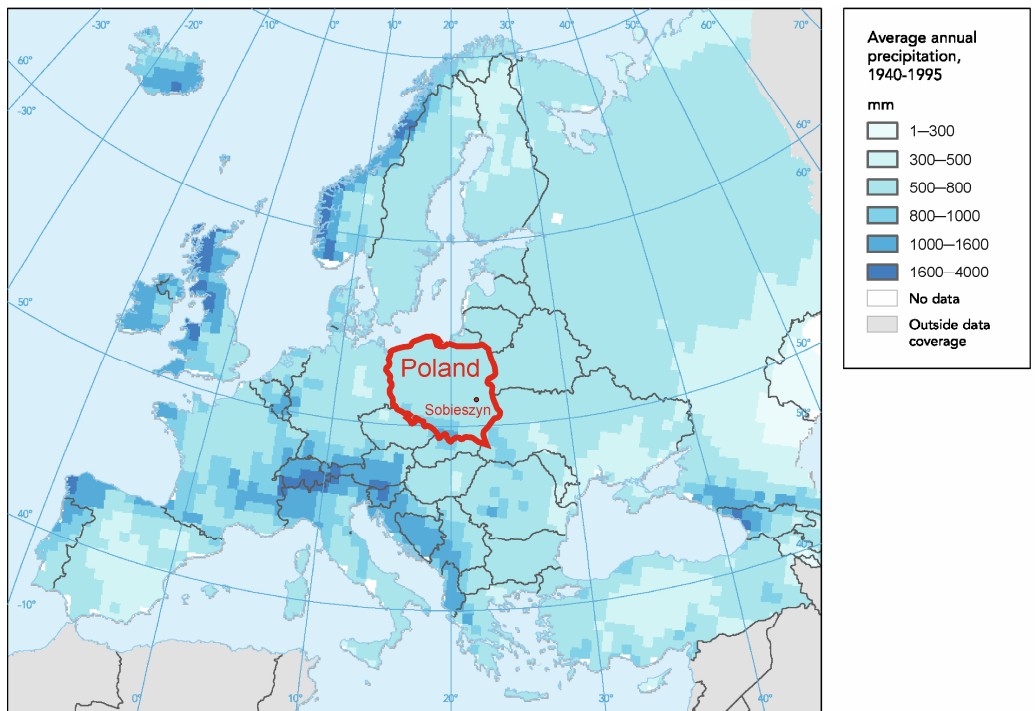

**Figure 1.** Location of the study facility on the map of average annual precipitation in Europe (map drawn by the authors based on [29]).

## 2. Materials and Methods

### 2.1. Description of the One-Stage Constructed Wetland

The investigated CW is located in central Europe, in the south-east of Poland (Figure 1), in a temperate climate zone with transitional features. The transitions are a result of friction between moist maritime air masses flowing in from the west (from the Atlantic Ocean) and dry continental air masses flowing in from the east (from the vast territory of Asia). The

characteristic feature of such a climate is the occurrence of four seasons in the year with different temperatures and different levels of precipitation.

The investigated CW was built in 1995 near the School of Agriculture in Sobieszyn in southeastern Poland (geographic coordinates: 51°35′37″ N 22°09′45″ E). The treatment plant (Figure 2) consists of the following components: a two-chamber preliminary septic tank, a sewage pumping station, and four vertical-flow parallel reed beds with reeds that have a total area of 1287 m$^2$ [30]. The beds are made of several layers of different materials, one layer of fabric and drains. The surface layer, with a thickness of 0.2 m, is made of humus overburden. It is underlain by a layer of loose sand of the same thickness. A 1.2 mm thick filter cloth is lain directly under the sand. Beneath it, lies a 0.3 m layer of dolomite crushed stone that has fragment sizes of 16–32 mm. Below, is a layer of drains collecting sewage. The diameter of each drainage pipe is 100 mm. The lowest layer, with a thickness of 0.1 m, is sand, and a 1 mm thick PEHD geomembrane is lain directly below it (whose task is to isolate the deposit from the natural soil). The receiver of sewage flowing out of the facility is a ditch adjacent to the forest, which discharges it into the ground [31].

The average flow rate of the wastewater treatment plant was 18.2 m$^3$·d$^{-1}$ during the research period. A facility is used for the treatment of domestic wastewater.

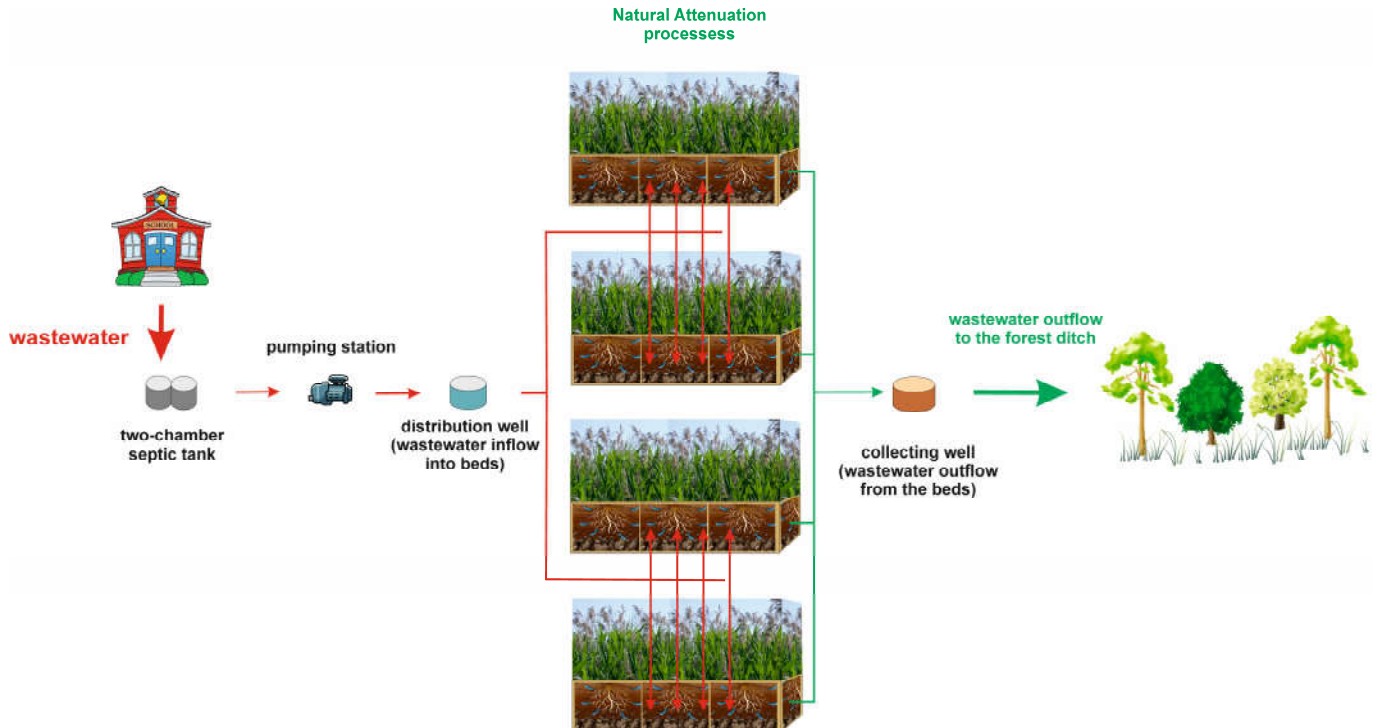

**Figure 2.** Technological scheme of the constructed wetland system in Sobieszyn (source: drawn by the authors).

*2.2. Description of the Database and Basic Analyses*

Physico-chemical analyses of the wastewater that is supplied to and discharged from the CW in Sobieszyn were carried out in the years 2001–2010. Precipitation and air temperature were measured at a weather station in Lublin, about 60 km away from the facility. The full database that was used, without any gaps in observations, was sufficient to present the results for the whole decade of 2001–2010, as well as monthly and yearly data. Additionally, we divided the meteorological observation data into the growing season, covering the period from 1 April to 31 October, and non-growing season, covering the period from 1 November to 31 March.

Treated wastewater from the analyzed facility was sampled for testing at regular intervals at four times a year (at the beginning of February, May, August, and November).

During the study period, 40 series of analyses were done. Wastewater samples were analyzed for their concentrations of total suspended solids (TSS), BOD5, COD, total nitrogen (TN), and total phosphorous (TP). Four hundred assays were performed using the wastewater samples to determine the composition of raw and treated wastewater. Samples were tested in the Water and Wastewater Testing Laboratory at the Department of Environmental Engineering and Geodesy, University of Life Sciences, Lublin. The wastewater samples were tested following the guidelines provided in "Reference methodologies for the analysis of wastewater samples" and the methodology described in the Polish Standards [32]. The methods we used to determine the values of the analyzed parameters are given below:

— The content of TSS was determined by the direct gravimetric method while using filters;
— $BOD_5$ was determined by the dilution method ($O_2$ concentration was measured using an Oxi 538 oxygen meter from WTW):
— $COD_{Cr}$ was determined by the dichromate method (COD was measured with an MPM 2010 photometer from WTW after a prior oxidation of a sample in a thermoreactor at 148 °C);
— The concentration of total nitrogen was determined with a PCspectro spectrophotometer from AQUALYTIC after the oxidation of a sample in a thermoreactor at 100 °C;
— The concentration of total phosphorus was determined with the MPM 2010 photometer from WTW after the oxidation of a sample in a thermoreactor at 120 °C.

The standard values of the examined variables are shown in Table 1.

**Table 1.** Standard values in the wastewater flowing from small treatment plants (<2000 PE), according to [32].

| Parameter | Standard Value in the Outflow [mg/L] |
|---|---|
| COD | 150 |
| $BOD_5$ | 40 |
| Total suspension | 50 |
| Total nitrogen | 30 |
| Total phosphorous | 5 |

Basic statistical analysis of the variation in the parameters observed was performed using Statistica software version 8.0. Precipitation data for the correlation analysis covered the 10 days preceding the collection of wastewater samples. The relationships between the parameters of the treated wastewater effluent were presented assuming that the temperature/precipitation values were independent variables, and the other parameters were dependent variables. The associations were determined using Pearson's linear correlation. The correlation coefficients were interpreted using a scale developed by Stanisz [33].

Since the tested facility covers an area of 1287 m$^2$, it was assumed that it may contribute to increasing the amount of inflow and dilution of wastewater flowing out of the treatment plant. Therefore, in this study, the relationship between the amount of monthly sums of atmospheric precipitation and the increase in the amount of wastewater flowing out of the treatment plant was determined.

### 2.3. Precipitation and Temperature in the Study Period

In the period covered by the analysis, both average annual air temperatures and annual precipitation totals were characterized by typical variation. As shown in Figure 3, the highest annual precipitation was recorded in 2011 (751 mm), and the lowest in 2003 (492 mm). In turn, the highest average annual temperature of 8.9 °C was recorded for 2008, and the lowest was 7.5 °C for 2010. The coefficient of variation calculated for the entire study period was 13% for precipitation and 6% for air temperature, which means the variation was low.

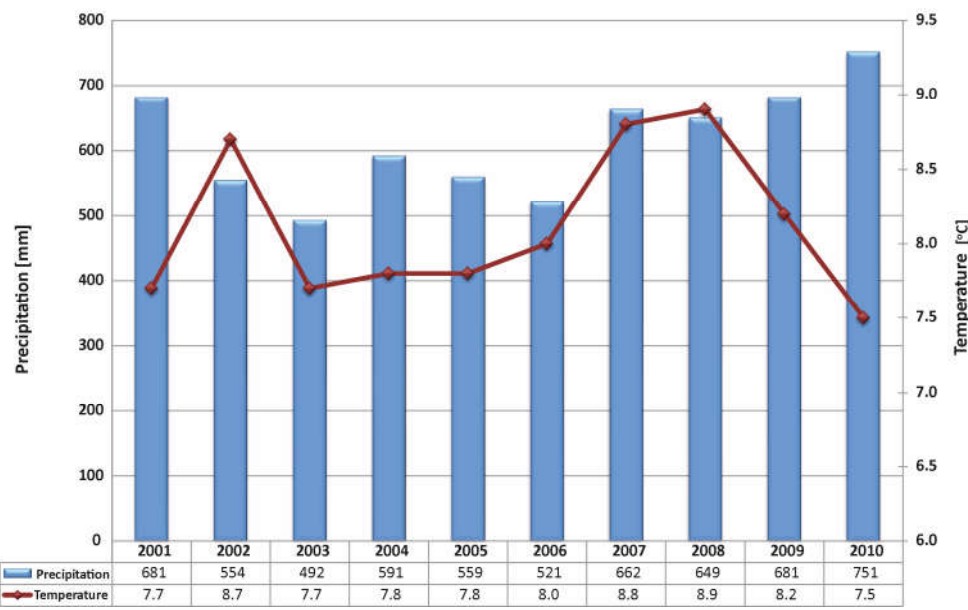

**Figure 3.** Mean annual precipitation and air temperature.

In accordance with the methodology described in Section 2.2, we decided to divide the meteorological observation data into two periods: a growing season and a non-growing season. Data grouped in this way enabled a further advanced analysis of the impact of precipitation and temperature on the concentration of pollutants in the effluent from the one-stage CW with subsurface vertical flow in Sobieszyn. This division was made to set apart the two periods that had extremely different weather conditions. We hypothesized that these conditions affected the efficiency of wastewater treatment in the investigated constructed wetland, and better results were expected for the growing season. The average annual precipitation and air temperatures by season (growing vs. non-growing) are shown in Figure 4.

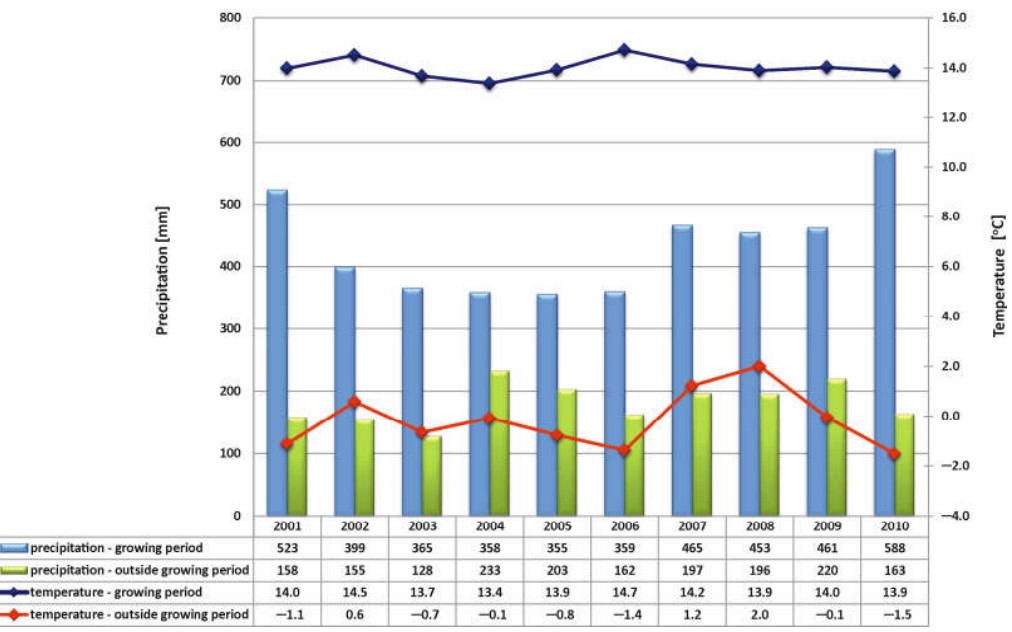

**Figure 4.** Mean annual values of precipitation and air temperature divided into the growing and non-growing season.

## 3. Results and Discussion

### 3.1. Wastewater Treatment Efficiencies during the Growing Season

The values of the test parameters at the inlet to and outlet from the CW in Sobieszyn for the growing season from April to October are given in Figure 5. Both the maximum values and means of the measurements indicate that the pollutant load in the wastewater at the outlet from the CW was significantly reduced compared to the inlet (Table 2). The analyzed indicators and parameters of both the influent and effluent wastewater were characterized by high variation. The exceptions were the content of nitrogen in both the influent and effluent wastewater, as well as the values of $BOD_5$ and COD at the outlet, which all showed moderate variation.

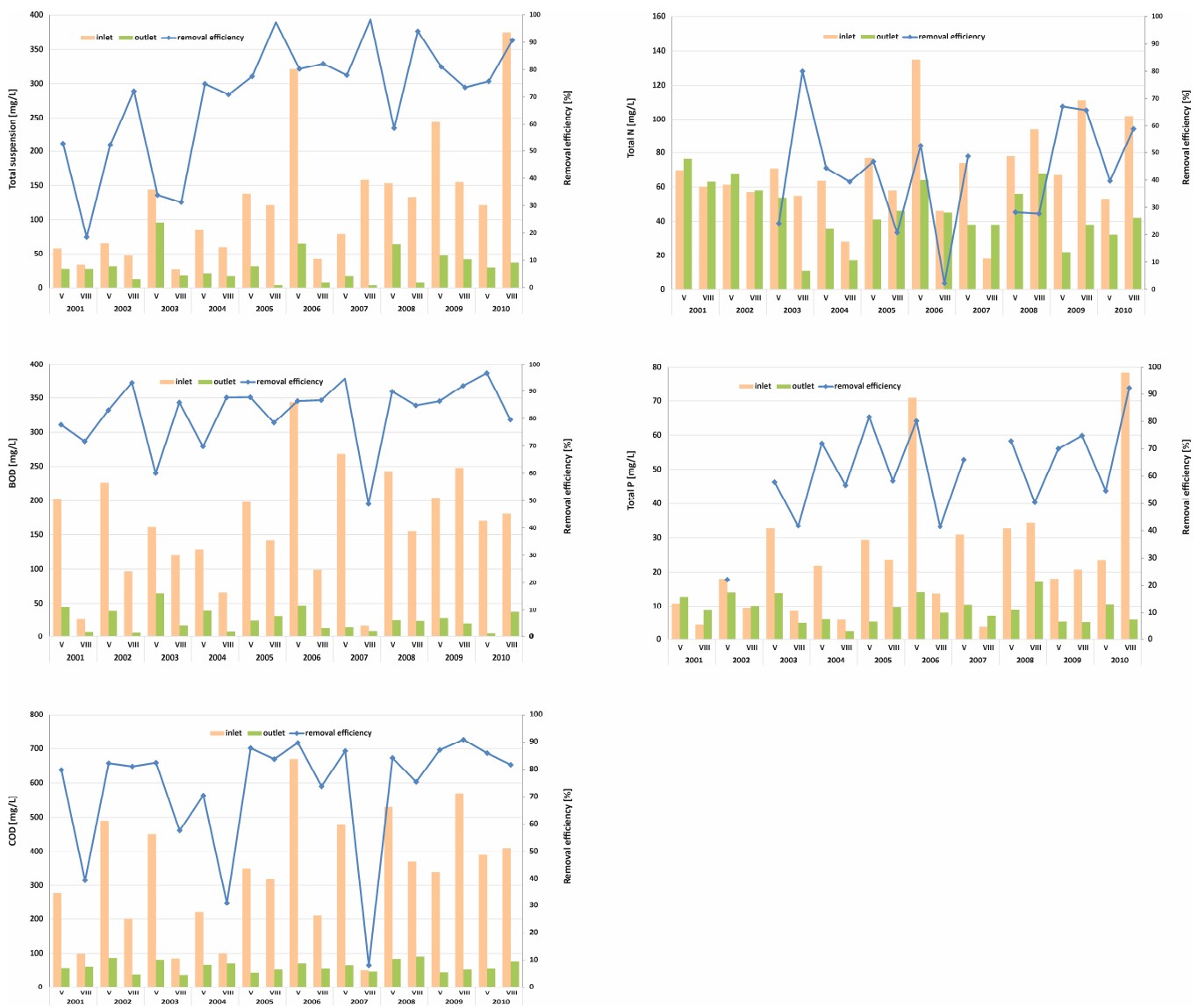

**Figure 5.** Influent and effluent values of the test parameters and pollutant removal efficiencies in the growing season.

**Table 2.** Statistical characteristics of the concentrations of pollutant parameters in the influent and effluent wastewater during the growing season.

| Parameter | Sampling Site | Min [mg/L] | Max [mg/L] | Mean [mg/L] | Standard Deviation [mg/L] | Coefficient of Variation [%] |
|---|---|---|---|---|---|---|
| TSS | inlet | 27 | 375 | 128.1 | 93.2 | 73 |
| | outlet | 4 | 96 | 30.8 | 23.3 | 76 |
| BOD$_5$ | inlet | 17 | 344 | 164.8 | 81.9 | 50 |
| | outlet | 6 | 64 | 25.1 | 16.1 | 64 |
| COD | inlet | 50 | 670 | 330.6 | 174.5 | 53 |
| | outlet | 36 | 91 | 61.0 | 16.3 | 27 |
| TN | inlet | 18 | 135 | 68.9 | 26.9 | 39 |
| | outlet | 11 | 77 | 45.6 | 17.7 | 39 |
| TP | inlet | 4 | 78 | 24.6 | 19.8 | 80 |
| | outlet | 3 | 17 | 9.1 | 3.8 | 42 |

In the growing season, the CW always operated efficiently with regard to the removal of TSS, BOD$_5$, and COD, and a significant reduction in the values of these indicators in the effluent were found compared to the influent (Figure 5). Our analysis showed that the efficiency of the investigated vertical CW beds in reducing the wastewater concentrations of organic pollutants, measured as BOD$_5$, COD, and TSS, was satisfactory. In addition, for the growing period, the mean value was as follows: 82% for BOD$_5$, 73% for COD, and 69% for TSS. As for the nutrients, the concentrations of phosphorus and nitrogen in the effluent were several times larger compared to the inflowing wastewater.

The efficiency rates (Figure 5) calculated for all the test parameters were characterized by a very high variation. For TSS, BOD$_5$, and COD, a significant reduction was observed throughout the study period with the following efficiency rates: TSS 18% to 100%, mean 69%; BOD$_5$ 49% to 97%, mean 82%; and COD 8% to 91%, mean 73%. These data demonstrate that the pollutant removal efficiency of the treatment plant was highly varied. The figures for the removal of phosphorus and nitrogen were even more diverse, since the CW was occasionally more of a reservoir that enriched the wastewater with nutrients than a treatment facility. The large variation in the pollutant removal efficiency of the treatment plant provided the rationale for the present study, which aims to find a factor that determines this variation. In the next part of this paper, we report on the evidence regarding whether air temperature or the amount of precipitation could be such factors.

### 3.2. Wastewater Treatment Efficiencies in the Non-Growing Season

The values of the test parameters recorded at the inlet and outlet of the CW in Sobieszyn in the non-growing season from November to March in the years 2001–2010 are given in Figure 6.

Both the maximum values and the means of the measurements indicate that the pollution load in wastewater that had passed through the CW was significantly lower compared with the influent wastewater (Table 3). Data analysis, however, points to a higher variation of the test parameters in the effluent compared to the influent, with the exception of TSS. The test parameters were characterized by high variation, both for influent and effluent wastewater. The only exceptions were the content of nitrogen recorded at the inlet and outlet of the CW and COD in the influent wastewater—both of which showed moderate variation—as well as BOD$_5$, whose effluent levels were characterized by very high variation.

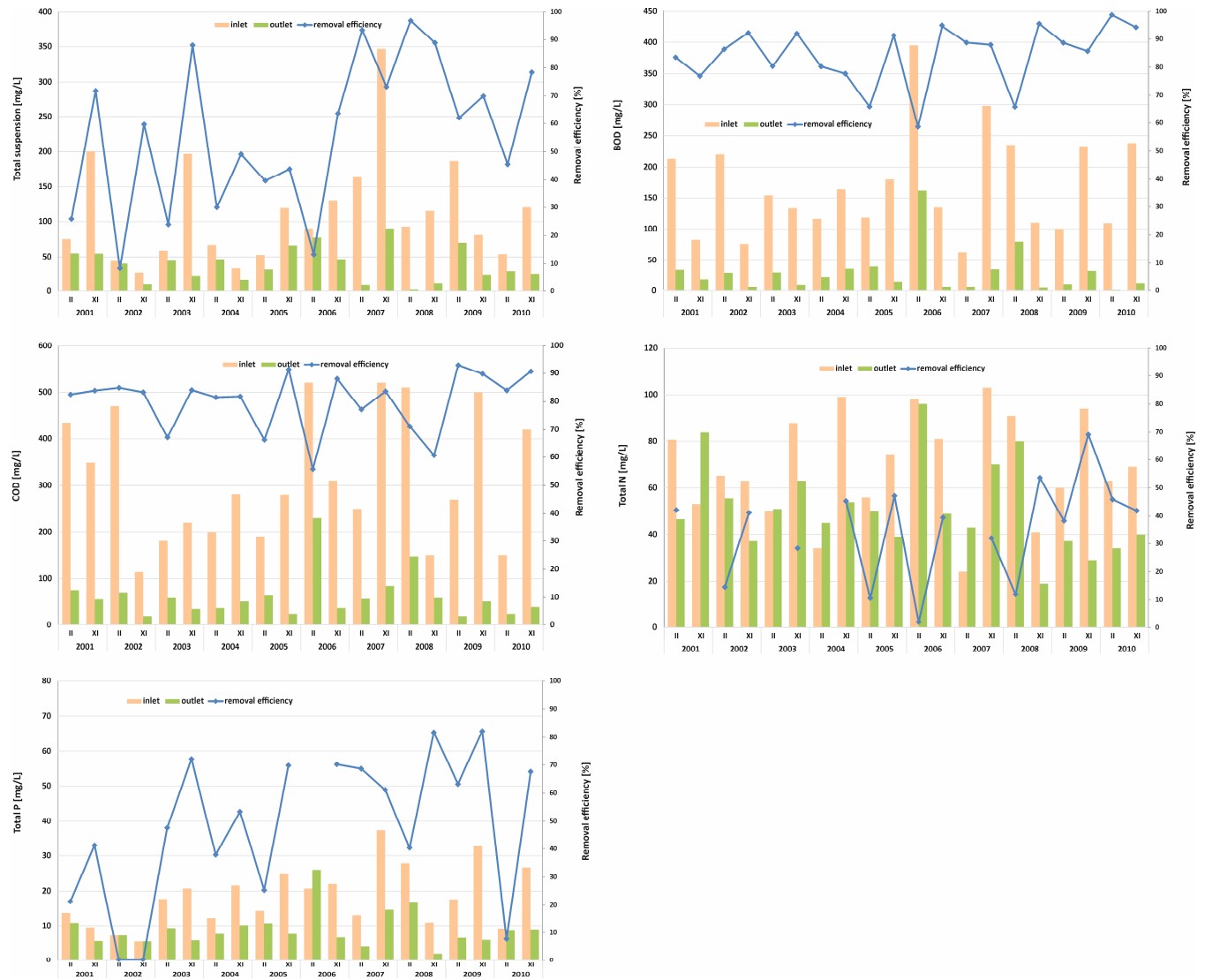

**Figure 6.** Influent and effluent values of the test parameters and pollutant removal efficiencies in the non-growing season.

In the analyzed non-growing season, similar to the growing season, the treatment plant always operated effectively in terms of the removal of TSS, BOD$_5$, and COD, whereby the values of these pollutants in the effluent compared to the influent wastewater was reduced (Figure 6). In regard to the biogenic substances, i.e., phosphorus and nitrogen, their concentrations in the effluent, on several occasions, were observed to be higher than those in the influent. Similarly, the efficiency rates (Figure 6) calculated for all the test parameters were characterized by a very high variation. The concentrations of TSS, BOD$_5$, and COD were significantly reduced throughout the study period with the respective efficiency rates being as follows: TSS 9% to 98%, mean 57%; BOD$_5$ 59% to 99%, mean 85%; and COD 56% to 93%, mean 80%. These data demonstrate that the pollutant removal efficiency of the CW in Sobieszyn was highly varied. The pollutant removal data for nitrogen and phosphorus, on the other hand, show that the CW occasionally functioned as a reservoir enriching the wastewater with nutrients rather than as a treatment plant.

**Table 3.** Statistical characteristics of the influent and effluent concentrations of pollutants in the non-growing season.

| Parameter | Sampling Site | Min [mg/L] | Max [mg/L] | Mean [mg/L] | Standard Deviation [mg/L] | Coefficient of Variation [%] |
|---|---|---|---|---|---|---|
| TSS | Influent | 27 | 347 | 112.7 | 76.8 | 68 |
| | Effluent | 2.2 | 90.0 | 38.6 | 24.6 | 64 |
| BOD$_5$ | Influent | 63 | 395 | 168.6 | 83.7 | 50 |
| | Effluent | 1 | 162 | 29.7 | 36 | 122 |
| COD | Influent | 114 | 520 | 315.9 | 138.9 | 44 |
| | Effluent | 19 | 230 | 62 | 49 | 79 |
| TN | Influent | 24 | 103 | 69.4 | 22.5 | 32 |
| | Effluent | 19 | 96 | 51.1 | 19.3 | 38 |
| TP | Influent | 5.5 | 37.4 | 18.3 | 8.7 | 47 |
| | Effluent | 2 | 26 | 9 | 5.2 | 58 |

*3.3. Impact of Precipitation and Temperature on the Concentration of Pollutants*

The relationship between the amount of precipitation or temperature (independent variable) and the amount of pollutants in the treated wastewater (dependent variable) was determined using Pearson's linear correlation. The correlation coefficients were considered significant at $p < 0.05$. The exploration procedure in the Statistica program, i.e., the Shapiro–Wilk test, was used to verify the normality of the distribution of the analyzed data. The results are shown in Table 4. Significance values greater than 0.05 indicate a normal distribution.

**Table 4.** The Shapiro–Wilk Tests for the analyzed parameters.

| Parameter | | Shapiro–Wilk Statistica "W" | Statistical Significance "*p*" |
|---|---|---|---|
| Temperature | | 0.906 | 0.054 |
| Precipitation | | 0.951 | 0.385 |
| TSS | | 0.890 | 0.269 |
| BOD$_5$ | Growing period | 0.932 | 0.170 |
| COD | | 0.963 | 0.613 |
| TN | | 0.973 | 0.816 |
| TP | | 0.960 | 0.540 |
| Temperature | | 0.935 | 0.195 |
| Precipitation | | 0.857 | 0.007 |
| TSS | | 0.961 | 0.558 |
| BOD$_5$ | Outgrowing period | 0.655 | 0.001 |
| COD | | 0.718 | 0.001 |
| TN | | 0.943 | 0.268 |
| TP | | 0.819 | 0.002 |

Significance values lower than 0.05 indicated that, for the outgrowing period, the precipitation, BOD, COD, and total phosphorous distributions differed significantly from the normal distribution. The other parameters were normally distributed. In order to restore the data to a normal distribution, it was decided to utilize the Box–Cox transform instead.

Selected correlation results for the analyzed parameters of the treated wastewater samples collected in the growing and non-growing period, in relation to the precipitation or temperature for the 10 days preceding the sampling, are shown in Table 5.

**Table 5.** Correlation matrix of the effect of precipitation and air temperature on the concentrations of the parameters measured in the treated wastewater.

| Factor | Growing Season | | | | | Non-Growing Season | | | | |
|---|---|---|---|---|---|---|---|---|---|---|
| | TSS | BOD$_5$ | COD | TN | TP | TSS | BOD$_5$ | COD | TN | TP |
| Precipitation | −0.31 * | −0.32 * | −0.02 | 0.04 | −0.14 | −0.59 * | −0.19 | −0.17 | −0.35 * | −0.38 * |
| Temperature | −0.43 * | −0.41 * | −0.09 | −0.03 | −0.08 | −0.37 * | −0.39 * | −0.34 * | −0.26 | −0.44 * |

(*) significance.

Pearson's correlation coefficients assumes values ranging from −0.03 for COD and precipitation in the growing season, to −0.59 for TSS and precipitation in the non-growing season.

The results of the correlation analysis that were interpreted using the scale proposed by Stanisz [33], showed that the only statistically significant relationship (a high correlation) was that between precipitation and TSS in the non-growing season (a correlation coefficient of 0.59). Moderate correlations were found between the precipitation and TSS and BOD$_5$ in the growing season, and between precipitation and TN and TP in the non-growing season. Similarly, moderate correlations were recorded between air temperature and TSS and BOD$_5$ in the growing season, as well as between air temperature and TSS, BOD$_5$, COD, and TP in the non-growing period. The correlation coefficients for all these relationships were higher than 0.3, which means the relationships were practically significant. The associations observed in the growing season are shown in Figure 7.

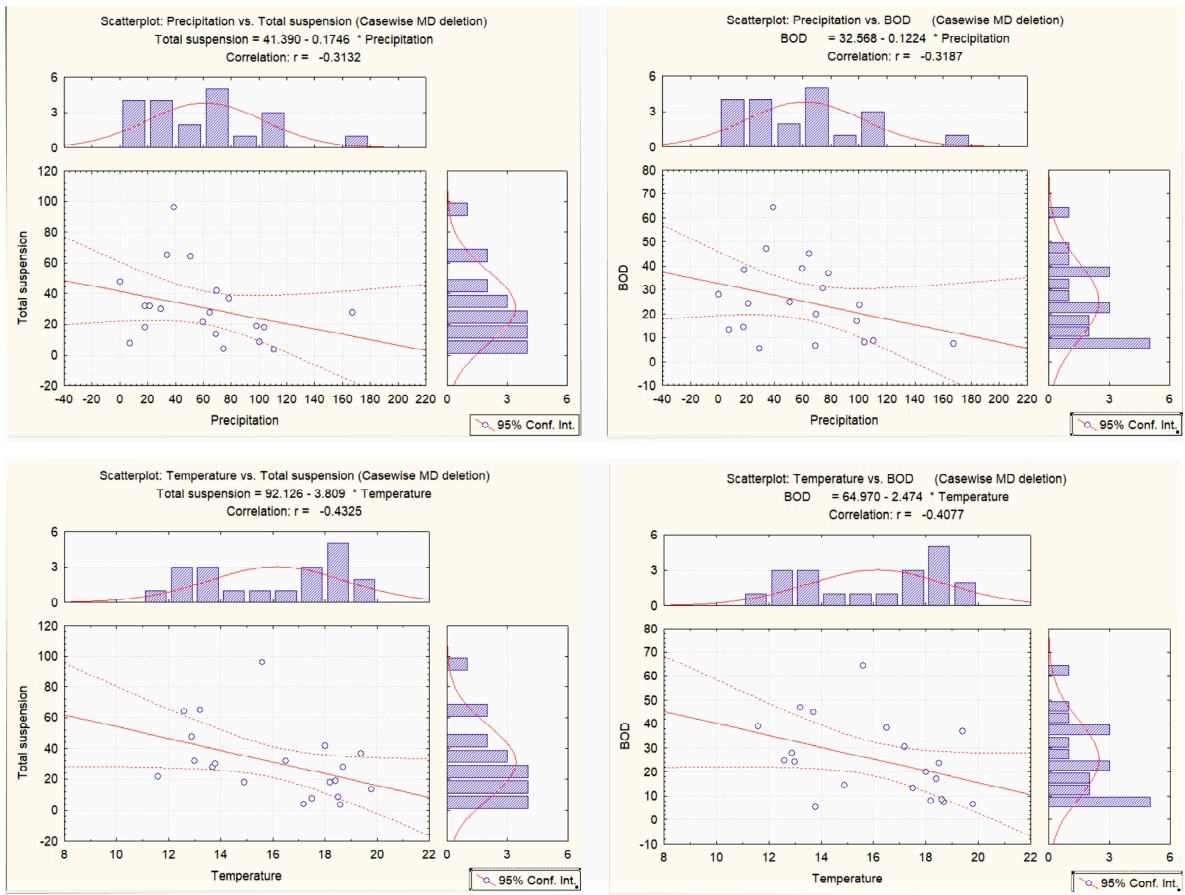

**Figure 7.** Graphs showing the moderate correlations observed in the growing season.

All the correlations listed above, and shown in Figure 7, were negative linear relationships, which means that, as air temperature/precipitation increased, the values of the dependent variables measured in the effluent decreased. The relationships observed for the growing season, described by the formulas given in Figure 7, indicate the following:

- With an increase in precipitation, the contents of TSS and $BOD_5$ decreased. A 10 mm rise in precipitation caused a drop in TSS by approx. 1.7 mg/L and a drop in $BOD_5$ by approx. 1.1 mg/L;
- With an increase in temperature, the contents of TSS and $BOD_5$ decreased. A 1 °C rise in temperature caused a drop in TSS by approx. 3.8 mg/L and a drop in $BOD_5$ by approx. 2.4 mg/L.

The relationships observed in the non-growing season are shown in Figure 8.

The correlations observed in the non-growing season, as shown in Figure 8, were negative linear relationships, which means that an increase in air temperature/precipitation was accompanied by a decrease in the values of the dependent parameters in the effluent. The relationships observed for the non-growing season, described by the formulas given in Figure 8, indicate the following:

- With an increase in precipitation, the contents of TSS, TN, and TP decreased. A 10 mm rise in precipitation caused a decrease in TSS content by approx. 5.3 mg/L, a decrease in TN content by approx. 2.5 mg/L, and a decrease in TP content by approx. 0.7 mg/L;
- With an increase in temperature, the contents of TSS, $BOD_5$, COD, and TP decreased. A 1 °C rise in temperature caused decreases in TSS by approx. 2.5 mg/L, in $BOD_5$ by approx. 3.8 mg/L, in COD by approx. 4.5 mg/L, and in TP by approx. 0.6 mg/L.

In all the cases, the correlation coefficients took a negative value, meaning that the mean values of the observed parameters dropped with an increase in precipitation. This stands to reason, since—according to literature reports [33]—rainwater contains only trace amounts of organic and biogenic pollutants, and so an increase in precipitation leads to the dilution of pollutant concentrations in the wastewater that is discharged from a treatment plant—thus leading to an improvement in its pollutant removal efficiency.

When taking into account the monthly sums of atmospheric precipitation during the research period, it can be concluded that precipitation had a large impact on the amount of wastewater outflowing from the studied constructed wetland system (Figure 9). For example, with 20 mm of rainfall per month, 25 $m^3$ of water fell on the four CW beds with a total area of 1287 $m^2$, which accounted for 5% of the average daily amount of wastewater flowing into the studied treatment plant. However, with the maximum monthly rainfall of 188 mm, which was observed in August 2006, almost 242 $m^3$ of water fell on the discussed facility within a month, which constituted about 45% of the average daily amount of inflowing wastewater. This probably had a large impact on decreasing the pollutant concentration in the outflowing wastewater.

In a study by Jóźwiakowski [31], the average share of rainwater in the total hydraulic load of CW beds in two single-stage and two hybrid CWs ranged from 13 to 33%. A similar hydraulic load with rainwater (a mean of 21%) was reported by Kuczewski and Paluch [34] for a CW located in Szewce in southwestern Poland. In a study conducted in Glaslough, Ireland, the average percentage of rainwater in the total hydraulic load of a 3.25 ha CW, serving a population equivalent of 800, was 55.8% [35]. These data indicate that rainwater can considerably dilute the wastewater treated in CWs, thus improving the efficiency of treatment.

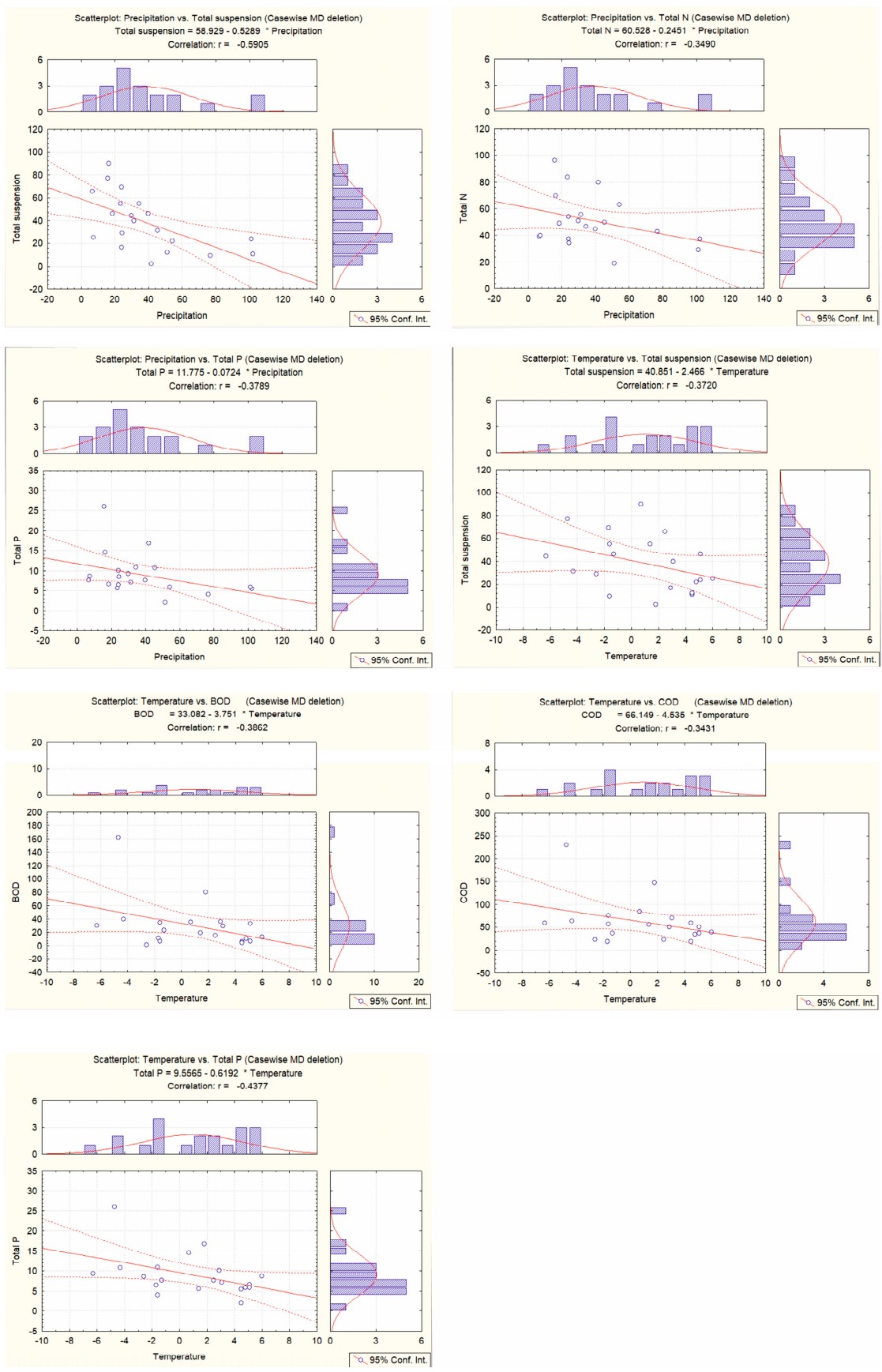

**Figure 8.** Graphs for the moderate correlations observed in the non-growing season.

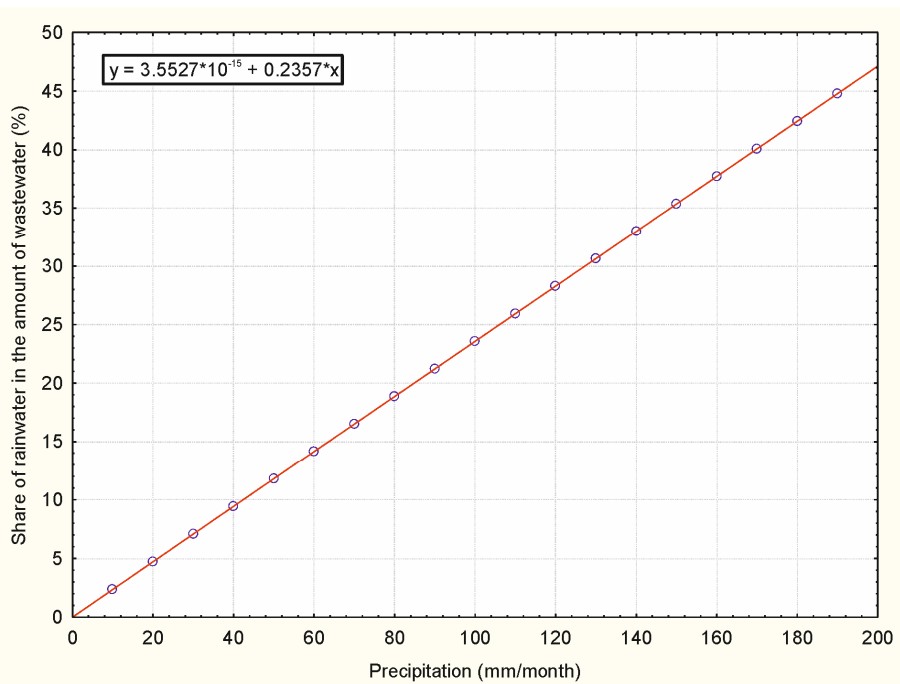

**Figure 9.** The impact of the amount of precipitation on the increase in the outflow of wastewater from the studied constructed wetland system.

## 4. Conclusions

Our analysis showed that the efficiency of the investigated vertical-flow CW beds in reducing the wastewater concentrations of organic pollutants—measured as $BOD_5$, COD and TSS—was satisfactory. For the growing period, the results were the following: 82% for $BOD_5$, 73% for COD, and 69% for TSS. For the non-growing period, the results were as follows: 85% for $BOD_5$, 80% for COD, and 57% for TSS. In the case of biogenic parameters, i.e., TN and TP, the processes taking place in the vertical-flow beds did not guarantee a reduction in the concentrations of these parameters to the permissible level laid down in the legal provisions applicable to treatment plants of this size. It was found that, in the non-growing season, the variation in the efficiency of removing organic pollutants and reducing the concentration of TP was influenced by fluctuations in air temperature. The rainwater supplying the CW beds reduced TSS and the biogenic parameters (TN and TP) in the non-growing season. In the growing season, variations in air temperature and precipitation only affected TSS and $BOD_5$. With an increase in precipitation, the contents of TSS, TN, and TP in the outflow decreased. A 10 mm rise in precipitation caused a decrease in TSS content by approx. 5.3 mg/L, a decrease in TN content by approx. 2.5 mg/L, and a decrease in TP content by approx. 0.7 mg/L. When there was an increase in temperature, the contents of TSS, $BOD_5$, COD, and TP in the outflow also decreased. A 1 °C rise in temperature caused decreases in TSS by approx. 2.5 mg/L, in $BOD_5$ by approx. 3.8 mg/L, in COD by approx. 4.5 mg/L, and in TP by approx. 0.6 mg/L.

The presented research results are part of the goal of sustainable wastewater management as they indicate an alternative to wastewater treatment for the popularly used treatment plants with activated sludge technology, even though reverse osmosis technology is promising in terms of increase the efficiency of pollutant removal from wastewater [36]. The advantages of the constructed wetland treatment method include low operating costs, water (sewage) retention in the hydrophyte bed, and a uniformity in the hydrophyte bed with the surrounding area. The listed advantages of the constructed wetlands represent the worldwide discussion regarding the need to retain water and minimize energy consumption. We believe that the results of the research will find readers, especially among

people and institutions that make decisions regarding the construction or modernization of wastewater treatment plants.

As a practical recommendation, we propose that the investigated type of technological system, which consists of a septic tank and vertical-flow CW beds, should be used in locations where legal provisions require a high efficiency in treatment with regard to the basic parameters: $BOD_5$, COD, and TSS. If the requirements also apply to biogenic parameters, then hybrid systems (i.e., combining vertical-flow and horizontal-flow beds) should be used.

**Author Contributions:** Conceptualization, A.O., K.J. and P.B.; methodology, K.J. and P.B.; formal analysis, A.O., K.J., and P.B.; investigation, A.O., K.J., W.J. and P.B.; resources, A.O., K.J., W.J., J.R. and P.B.; data curation, K.J. and P.B.; writing—original draft preparation, A.O., K.J. and P.B.; writing—review and editing, A.O, K.J., W.J., J.R. and P.B..; visualization, A.O.; supervision, K.J., W.J. and P.B. All authors have read and agreed to the published version of the manuscript.

**Funding:** This paper was written on the basis of the following research projects funded by the Scientific Research Committee and the Ministry of Science and Higher Education of Poland: Analysis of the functioning of small wastewater treatment plants in rural areas and attempts to increase their effectiveness in the aspect of environmental protection - (contract no. 3 P06S 058 23, 2002–04); Optimization of pollution removal processes in small wastewater treatment plants under model and field conditions (contract no. N N523 3495 33, 2007–10).

**Institutional Review Board Statement:** Not applicable.

**Informed Consent Statement:** Not applicable.

**Data Availability Statement:** The data are not publicly available due to privacy restrictions.

**Conflicts of Interest:** The authors declare no conflict of interest.

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
