# Peer review of "Impact of Climate Conditions on Pollutant Concentrations in the Effluent from a One-Stage Constructed Wetland: A Case Study"

_sustainability, doi:10.3390/su151713173_

Round 1

Reviewer 1 Report

Moderate changes to the level of English would be beneficial

Author Response

Authors would like to thank for reviewing the manuscript. We greatly appreciate the Reviewer for his comments and suggestions. We have worked so hard last week on the manuscript and we have tried to do our best to respond to the points raised and make the paper acceptable for publication. All Reviewers have brought up some good points and we appreciate the opportunity to clarify our research idea. We have checked all the comments provided by the Reviewers and have made the necessary changes according to their indications.

Please find response to your concerns shown below. All changes were marked in colour in the manuscript (track-changes version). We hope that responses will be satisfactory. We strongly believe that currently the paper could be acceptable for publication.

Please, see the attachment with detailed response, manuscript as "track-changes" and manuscript as "clean-version".

Reviewer 2 Report

This paper investigated influence of precipitation and air temperature on the efficiency of pollutant removal processes and effluent pollutant concentrations in a one-stage constructed wetland with subsurface vertical flow. The author studied an on-site constructed wetland system with Phragmites australis used for the treatment of domestic wastewater. Although manuscript is well written, further improvement is necessary. Specific comments are shown as follows.

1. Line 25-26, the reason why rainfall diluted the concentration of wastewater could improve the efficiency of the wastewater treatment plant should be further explained. The related part in manuscript also should be supplied.

2. The introduction should be rearranged because it contains plenty of common knowledge, some of the recent references about reclaimed water should be cited (Journal of Cleaner Production 2023, 417, 138029; Journal of Molecular Liquids 2023, 380, 121704; Membranes 2023, 13, 355). In addition, the contrast from other research about CW should be described in this part.

Also, there are some minor problems.

1. It could be more convincible to add some specific data in Abstract such as removal efficiency.

2. Figure 1 should be placed in section 2.1 and combined with Figure 2.

3. Section 3.1 is not the authors’ research goal in this manuscript rather than a basic condition in study area. This part should be described in supplementary materials.

4. Error bar should be given in the Figure 6.

5. Precondition of Pearson’s correlation is that variables should conform normal distribution. Please check it.

6. Table 3, are all the numbers in correlation matrix significant (p<0.05)? The symbol of significance (*) should be marked.

7. Line 325, What is the full name of CV? The full name of abbreviation should be given at the first time.

Minor editing of English language required

Author Response

(The authors gave the same response as above.)

Reviewer 3 Report

Greetings and courtesy. It is a good topic, but it should be improved a little, especially the language of this article and the references should be updated. The language of the article needs general editing.

Author Response

(The authors gave the same response as above.)

Reviewer 4 Report

-          The abstract contains results that even people unrelated to environmental knowledge can express by default. It would have been better if it had been expressed in a more scientific way, for example, which pollutants were analyzed and these results were obtained. Or it would be better if the concentration of pollutants and different variables were expressed in different temperature and humidity conditions.

-          The introduction contains additional and unrelated material to the study. The last part of the introduction is related to the methods and materials section.

-          At the beginning of the introduction, it would have been better to show the amount of the studied pollutants in the investigated wastewater in a table before starting the study. (total suspended solids (TSS), BOD5, COD, total nitrogen (TN) and total phosphorous (TP)).

-          - Figures 6 and 7 are not clear.

-          It would have been better if the standard values of the examined variables were shown in the tables as well.

-          The conclusion section should be corrected. It is better to mention the working method, the importance of this study and the results obtained

Author Response

(The authors gave the same response as above.)

Round 2

Reviewer 3 Report

The problems have been solved.

Reviewer 4 Report

However, there are still problems in the abstract, introduction, and conclusion. But this article format is acceptable